# Regional scale shear wave velocity profiles for ground response analyses and uncertainties evaluations – the Piedmont Region (NW Italy) Database.

Cesare Comina [1], Guido Maria Adinolfi [1*], Carlo Bertok [1], Andrea Bertea [2], Vittorio Giraud[2], Pierluigi Pieruccini[1]

[1] Dipartimento di Scienze della Terra, Università degli studi di Torino, Torino, Italy.
[2] Seismic Sector, Piedmont Region, Pinerolo, Italy.

*Correspondence to*: Guido Maria Adinolfi (guidomaria.adinolfi@unito.it)

**Abstract.** Ground response analyses using a statistically representative sample of soil and rock profiles are typically used to estimate earthquake ground motions and, in turn, the seismic hazard of a particular area of study. With this aim shear wave velocity (Vs) properties of the profiles are of paramount importance, given that uncertainty in this parameter play a major role in ground motion prediction and in its variability. Usually, stochastic procedures are adopted to model this uncertainty, and several stochastic approaches have been developed. These approaches should be however calibrated on detailed geological-geomorphological information and specific Vs profiles databases. Within this context the present paper is aimed to provide a new extensive database of Vs profiles over the Piedmont Region (NW Italy). These data are obtained through a specific workflow developed for their evaluation at the regional scale merging the information of specific geological-geomorphological modelling and devoted geophysical data collection. The obtained database (https://doi.org/10.5281/zenodo.13685087) could be used as the basis of Vs randomization approaches also in different geological contexts and results from the specific data analyses performed could be adopted as reference for similar materials in analogous geological contexts.

## 1 Introduction

The prediction of earthquake ground motions, and consequent seismic hazard of a specific area of study, is usually based on ground response evaluations of a statistical representative sample of possible soil and rock profiles (i.e. seismo-stratigraphical profiles) in the area (Pieruccini et al., 2022). As an example, "amplification abacuses" are widely diffused simplified tools for the quantification of local stratigraphic amplifications of the seismic ground motion over large areas, i.e. Regions. These evaluations are therefore the result of a compromise between generalization and specialization (Peruzzi et al., 2016) and several approaches have been adopted in the past for their formulation (e.g. Pagani et al., 2006). One of the challenging aspects of these kind of analyses is the definition of a Geological-Geomorphological Model (GGM) at regional-scale build on purpose for the assessment of shear wave velocity (Vs) properties of the seismo-stratigraphical profiles, their spatial distribution and

related geological variability. Particularly, for regional-scale studies the uncertainty in the Vs profiles definition is considered as the main source of uncertainty in ground response evaluations (Toro, 2022) and must be therefore considered with devoted attention.

Usually, stochastic procedures are adopted to model this uncertainty. Several stochastic approaches have been developed

through the years with parameters that should be however calibrated on specific GGMs and Vs profiles databases (e.g. Toro, 2005; Shi and Asimaki, 2018; Passeri et al., 2020). Indeed, older and widely used formulations of these approaches, e.g. Toro (1995), provided parameters calibrated on California profile data to be used elsewhere. New generic and site-specific stochastic Vs models should be therefore developed using specific databases or increased databases number and population, together with insights gained in the practical use of these models.

Several research efforts have focused on constructing and analysing Vs databases for different purposes, including: 1) developing site investigation guidelines, as demonstrated by EPRI (1993), with a database containing over 350 Vs profiles (mainly within United States); 2) managing uncertainties, as in Toro (1995), who compiled a database containing 745 Vs profiles from the PEA (Pacific Earthquake Analysis) database for the development of a geostatistical model; 3) addressing data gaps, as shown by Stewart et al. (2014), creating a Vs database for Greece using open-source data to extrapolate Vs,z (the

harmonic average shear-wave velocity profile down to depth z); 4) creating empirical correlations, as in Passeri et al. (2021), developing a database of 71 Vs profiles for statistical analysis and model calibration; 5) validating simplified methods, as in Aimar et al. (2019), using a Vs database to validate soil amplification factors in the Italian building code NTC (2018); 6) assessing measurement uncertainty, as in Moss (2008) and Comina et al. (2011), with smaller databases of 30 and 10 Vs profiles, respectively; and 7) supporting ground motion studies, as in Wang et al. (2019), establishing the United States

Community Vs database for ground motion and site response analysis.

In the present paper a new methodological workflow for the assessment of a GGM and related Vs profiles distribution at regional scale is presented, which is used to develop a new geological and geophysical database. Using existing datasets, implemented and validated on purpose, a new geographic database for ground response at regional scale was developed. The methodological workflow is tested over Piedmont Region in North-West Italy. This Region includes: the Alpine Mountain

environment; the Foreland Hilly landscape both with different bedrock and cover terrains typologies and thicknesses; the Po River plain, and secondary alluvial plains, with thick Quaternary deposits overlying at depth different bedrocks. Following the proposed workflow, we assessed a new Vs database performing a quality control of all the available datasets and producing additional information in areas not covered or poorly covered by data.

Therefore, this paper has three main aims: i) provide a new, extensive (i.e. containing more than 1000 profiles), database of

Vs profiles to be used as the basis of randomization approaches also in different geological contexts; ii) provide median properties of the different investigated geological-geomorphological domains to be adopted as reference for similar materials in analogous geological contexts; iii) provide a workflow to be adopted for the evaluation of Vs profiles distribution at the regional scale by merging geological-geomorphological information and specific geophysical data collection.

## 2 Geological-Geomorphological Model

The assessment of a Geological-Geomorphological Model (GGM) at regional scale is the first step of the proposed procedure. In order to get an updated synthesis of the geological knowledge the Geological Map of Piedmont Region (1:250000 scale) (Piana et al., 2017a, b, 2020) has been used. This map is available as open-source Geodatabase (Geoportale-ARPA) and therefore can be used for on purpose reclassification.

Three main reclassification levels of the geological information are needed to obtain a GGM consistent with the seismic
perspective:

1) Reclassification of the outcropping and subsoil units as Geological Bedrock and Cover Terrains. To simplify the model, Cover Terrains are usually those of Quaternary age, whereas Geological Bedrock can be considered as pre-Quaternary (Pieruccini et al., 2022);

2) Classification of each Geological Bedrock and Cover Terrain according to their main geotechnical properties
(Romagnoli et al., 2022; Gaudiosi et al., 2023);

3) Gathering of the original formations units into different Geological-Geomorphological Domains (GGDs) based on their stratigraphic-sedimentological characteristics and the geomorphological context of outcrop, including the range of thicknesses of expected subsoil characteristics.

The Geological and Geomorphological setting of the Region is the result of a complex geodynamical evolution that since the
Mesozoic led to the formation of two passive continental margins and two oceanic zones. The collision of the two margins after the Eocene is the beginning of the Alpine-Apennines orogenesis, characterised by complex metamorphic, magmatic and sedimentary processes (Piana et al., 2017b and references therein). The definitive emersion of the area is marked by the Middle Pliocene-Quaternary continental successions and the present-day landscape, the consequence of this complex evolution, can be subdivided into four main Landscape Systems, or physiographic units.

1) Mountain Ridges
   a. Alpine Ridge, extending form SW to the NNE with an arcuate shape
   b. Apennine Ridge, E-W trending in the south-eastern part of the Region
2) Hills of Torino, Langhe and Monferrato
3) The Quaternary alluvial basins and valley systems of the Po River
a. The Po plain fed by rivers of alpine provenance
   b. The floodplains of the valleys crossing the Alpine Ridge
   c. The floodplains fed by the rivers crossing the Apennine Ridge
   d. The floodplains fed by the rivers crossing the Hills
4) The Quaternary frontal moraines and related systems of fluvio-glacial and fluvio-lacustrine sediments of the main
glacial amphitheatres extending in the Po plain at the mouth of the main alpine valleys (es. Ivrea, Rivoli).

The analysis and reclassification of the available geological and geomorphological database allowed the identification of 13 different Geological-Geomorphological Domains (GGD) as reported in Figure 1.

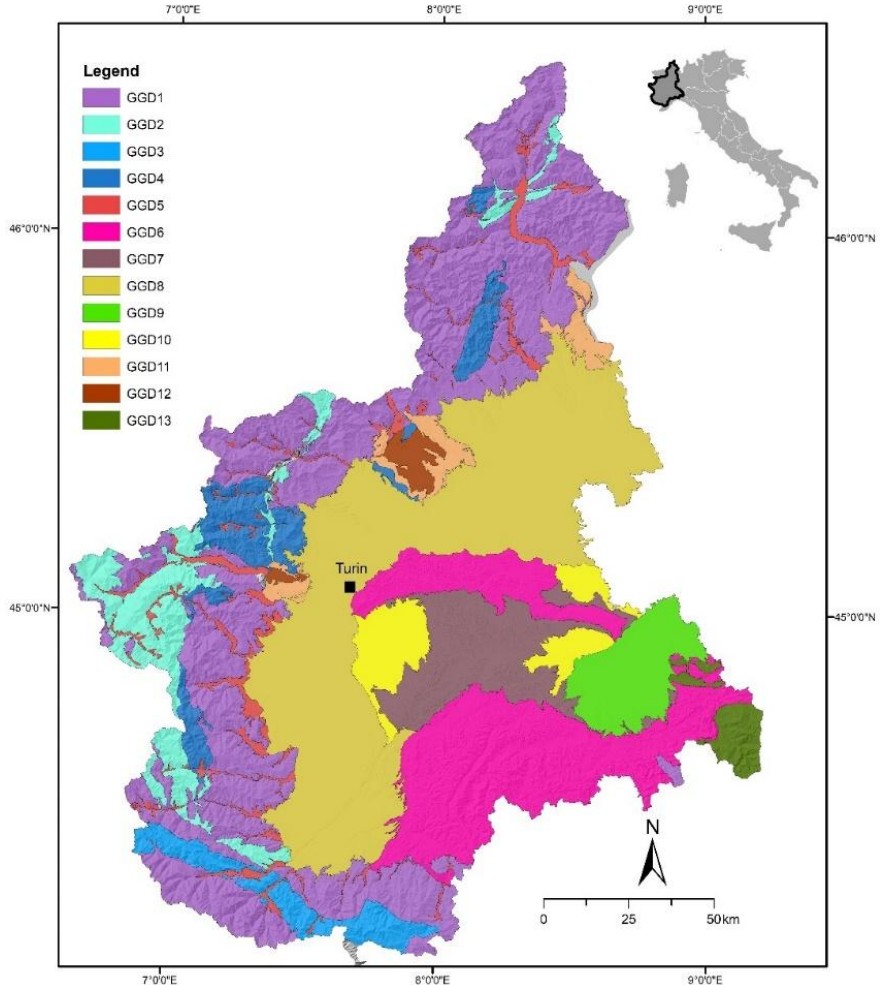

**Figure 1. Map of the Geological-Geomorphological Domains within the Piedmont Region.**

Each GGD is characterized by homogeneous Geological Bedrock typologies and potentially different litho-stratigraphic settings including Cover Terrains. The GGDs are related to (Table 1): a) the Alpine mountain chain with different bedrocks (GGD 1 to 4) including the main Alpine valleys (GGD 5); b) the foreland hilly landscape both with different bedrock and cover terrains typologies and thicknesses (GGD 6 and 7); c) the Po river plain, fed by Alpine rivers, with thick mostly coarse-grained Quaternary deposits overlying at depth different bedrocks (GGD 8); d) minor alluvial plains fed by rivers coming from
the Apennines and the foreland hills with thick mostly fine-grained Quaternary deposits overlying at depth different bedrocks (GGD 9 and 10); e) the moraine amphitheatres and the associated fluvio-glacial and lacustrine deposits (GGD 11 and 12); f) the complex successions belonging to the Ligurian Units (GGD 13).

**Table 1. The main characteristics of the GGDs**

| Physiographic Units | GGD | | GEOLOGICAL BEDROCK | COVER TERRAINS |
|---|---|---|---|---|
| **ALPINE RIDGE** | **1** | **METAMORPHIC AND MAGMATIC ALPS** | lapideous rocks and alternating litotypes | slope, alluvial, alluvial fan, glacial, fluvio glacial, mainly gravelly, packed, up to 100 m thick |
| | **2** | **OCEANIC SERPENTINITES** | lapideous to foliated rocks | |
| | **3** | **OCEANIC CALCESHISTS** | alternating lithotypes often weathered | |
| | **4** | **CARBONATIC ALPS** | lapideous rocks and alternating litotypes | |
| | **5** | **ALPINE RIVER VALLEYS** | | mainly unsorted alluvial gravels, pebbles and boulders, packed to cemented, up to 200 m thick |
| **FORELAND HILLS** | **6** | **OLIGO-MIOCENE BASINS** | alternating lithotypes, granular and cohesive | slope and alluvial unsorted gravels and sands up to 50 m thick |
| | **7** | **PLIOCENE BASINS** | mainly cohesive | slope and alluvial sands and silts up to 50 m thick |
| **QUATERNARY ALLUVIAL BASINS** | **8** | **PO RIVER PLAIN** | | alternances of dominant gravels, sands and silts, from loose to packed, up to 100 m thick |
| | **9** | **APENNINES RIVERS PLAIN** | | alluvial mainly gravels alternated to sands and silts loose to packed, up to 80 m thick |
| | **10** | **HILLS RIVERS PLAIN** | | alluvial mainly sands and silts, loose to packed, up to 80 m thick |
| **GLACIAL AMPHITHEATRES** | **11** | **MORAINE AMPHITHEATRES** | | mainly silts and clays loose or weakly packed, up to 40 m thick |
| | **12** | **FLUVIOGLACIAL AND LACUSTRINES** | | unsorted gravels, pebbles and boulders with sandy-silty matrix, loose to strongly packed, up to 200 m thick. |
| **APENNINE RIDGE** | **13** | **LIGURIAN UNITS** | alternating lithotypes often weathered | slope and alluvial unsorted gravels up to 50 m thick |

## 3 Data collection and QC

Once the GGDs were identified the available geotechnical and geophysical databases from Regional Authority's repositories

were used for the geological/geotechnical characterization and for their Vs parameterization. The main source of information was the Geotechnical Database of ARPA (Regional Agency for Environmental Protection of Piemonte Region) Piedmont (Geoportale-ARPA). This database contains several stratigraphic logs with various depths and quality. Attention was focused on the subset of about 3000 stratigraphic logs reaching at least 30 m in depth. Of these, more than 1000 logs, judged to be of higher quality, were consulted and are included in the presented database (Figure 2). Most of these logs reached the geological

bedrock and in these cases the bedrock depth was reported as information. Also, the prevalent properties of the Cover Terrains within the first 30 m were classified, when possible, according to the main textural and characteristics. The final data format used in the database for this information is reported in Table 2.

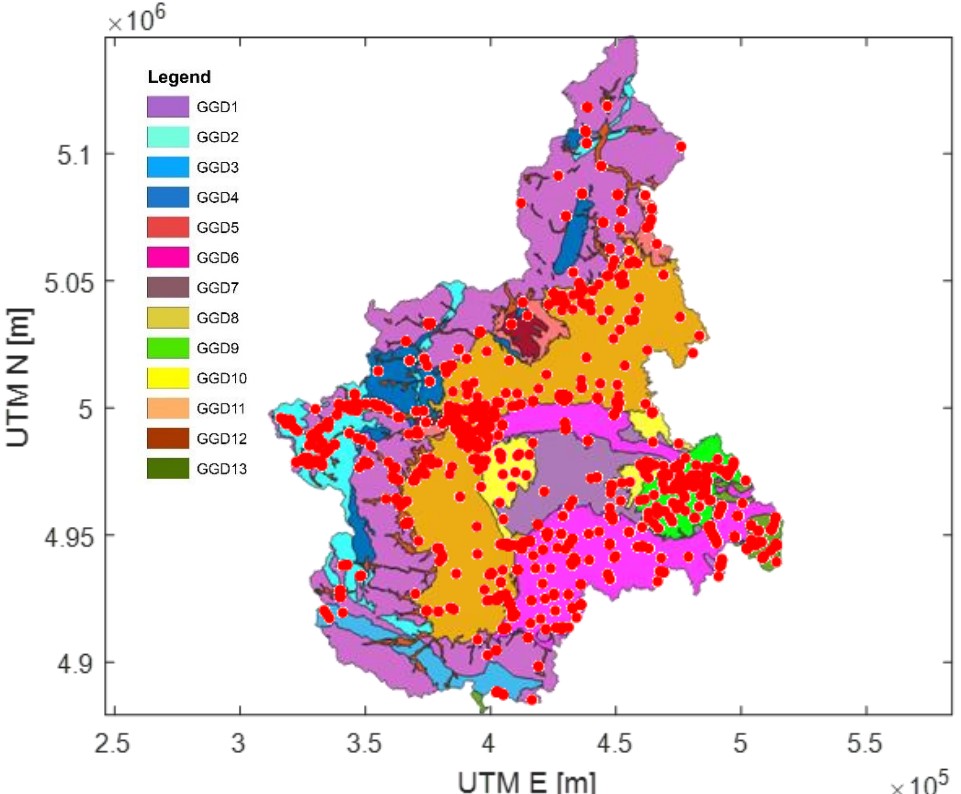

**Figure 2. Map of the GGDs' and the distribution of the stratigraphic logs (red dots) analysed (from Geoportale ARPA).**


**Table 2. Data format for the geotechnical information contained in the database. Data values for the bedrock depth and texture attributes derived by the available logs. UTM E and UTM N are the metric coordinate system used for their georeferentiation (WGS84 UTM32N)**

| | UTM E [m] | UTM N [m] | Geologic Bedrock depth [m] | Texture |
|---|---|---|---|---|
| **value** | - | - | Bedrock not reached or not clearly identified = 999 | C clay<br>G gravel<br>S sand<br>R rock<br>X not available |

With respect to the shear wave velocity properties the main source of data was the same Geoportale ARPA. The analysis of

this database allowed the assessment of about 2000 Vs profiles coming from both invasive and non-invasive tests. To fill the gap in the geographic data distribution we added more Vs profiles thanks to the collaboration with Techgea S.r.l., a leading geophysical private company that provided about 300 Vs profiles and by performing specific field tests or implementing specific information from literature data (about 50 Vs profiles).

Geophysical data underwent specific Quality Control (QC) in order to consider only reliable and state of the art information.

Particularly, the data deriving from Multichannel Analysis of Surface Waves (MASW) tests (the most widely diffused technique for Vs profile determination) underwent a specific QC consisting in checking: 1) consistency of the dispersion curve, that should present a clearly visible and continuous fundamental mode in the frequency band of interest; 2) when multiple dispersion modes occur, they should be well separable, well distinguishable and reliably interpretable independently; 3) a picking of the dispersion curve reliable and fitting with the spectral maxima of the seismogram transform used for the analysis;

4) inversion of the data leading to a synthetic dispersion curve having a good correspondence with the experimental data; 5) depth of the Vs profile compatible with the minimum frequencies observed in the analysis, i.e. investigation depth less than at least the maximum wavelength (preferably half the maximum wavelength); 6) Vs profile matching the minimum parametrization criterion, i.e. number of analysed layers compatible with the experimental information. The QC allowed the identification of about 1000 high-quality Vs profiles distributed over all the GGDs that were included in the final database

(Figure 3). The final data format used in the database for the Vs profiles information is reported in Table 3.

Both geotechnical and geophysical data distribution is influenced by the aims for which the different field tests were conducted. Particularly for geophysical data most information is inherited by the Seismic Microzonation studies, performed mainly on the Municipalities and settlements located within the Alpine valleys and at the border between the Alpine chain and the Po plain in the western sector of the Region, that is the area with higher Seismic Hazard. Data results therefore more concentrated

within and around the main urban settlements and most populated areas, that are the main targets for such type of studies (Figure 3).

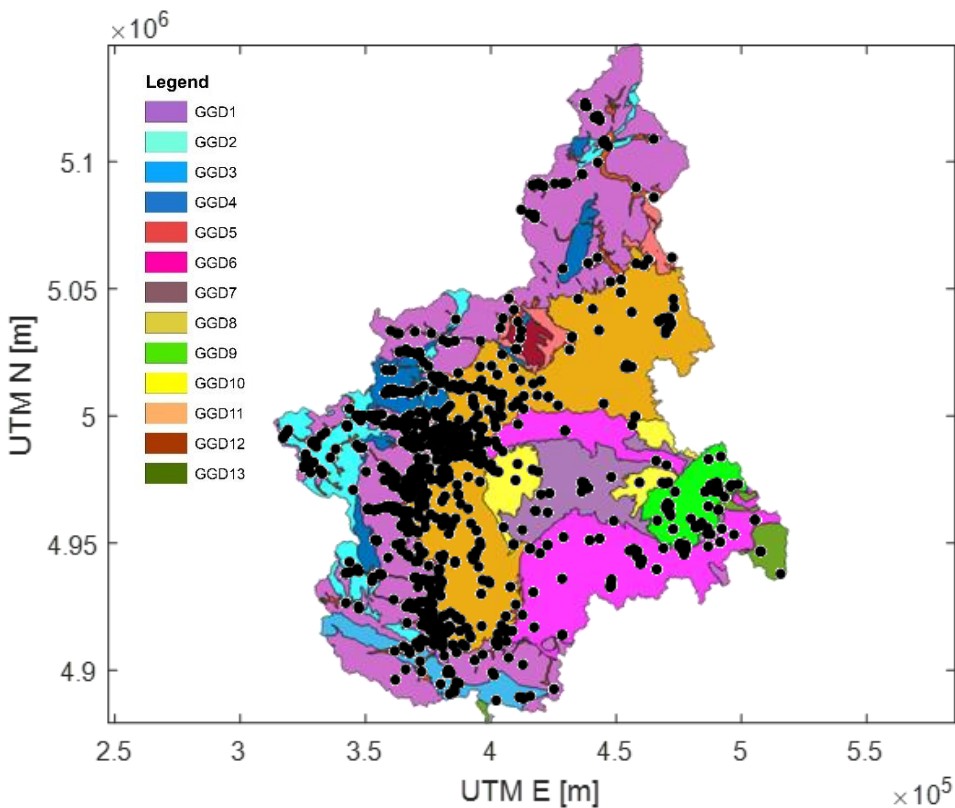

**Figure 3. Map of the GGDs' and the distribution of the Vs profiles (black dots) analysed (from Geoportale ARPA) after QC.**

**Table 3. Data format for the geophysical information contained in the database. Data values for the attributes of depth of seismic layers and Vs values. UTM E and UTM N are the metric coordinate system used for their georeferentiation (WGS84 UTM32N)**

| | UTM E [m] | UTM N [m] | Depth [m] | Vs [m/s] |
|---|---|---|---|---|
| **value** | - | - | Intended to be the layer interface depth | Intended to be the shear wave velocity above the layer depth |


Nevertheless, the obtained data allow to perform relevant analysis of the properties of the different materials characterizing each GGD. Particularly the final step of the workflow is the evaluation of specific Vs profile distribution within each GGD and their comparison among different GGDs. Also, plots of data properties distributions at the regional scale were produced in order to evaluate their variability at the regional scale.

## 4 Results and Discussion

Presented below are the statistical analyses that characterize the database introduced in this work. These analyses provide detailed insights into the structure and distribution of the data, offering a comprehensive understanding of the regional Vs profile distribution and its relevance to geophysical modelling. Also, suggested applications of the data made available within the paper are briefly discussed in the view of a wider use of the database within the interested community. Further data elaboration and specific detailed analyses are however outside the scopes of the paper which is intended to present the database by itself and leave to potential users the autonomy on eventual research on it.

The synthesis of the analysed parameters with respect to the Vs distribution for the different GGDs is reported in Table 4. Data distribution is not uniform among the GGDs due to the inhomogeneous geographical distribution of the data. The most populated GGDs are respectively GGD 8 (Po River plain) and GGD 5 (Alpine river valleys) whose results of the performed analysis are shown in Figures 4 and 5.

**Table 4. Analysed parameters with respect to the Vs distribution for the different GDs.**

| GGD | Number of Vs Profiles | Average Vs,h [m/s] | Vs profiles reaching the seismic bedrock | Seismic bedrock depth range [m] | Average bedrock Vs [m/s] |
|---|---|---|---|---|---|
| 1 | 68 | 390 | 45 | 0 - 40 | 1020 |
| 2 | 34 | 405 | 21 | 4.5 – 40 | 1020 |
| 3 | 11 | 470 | 8 | 4 – 25 | 1250 |
| 4 | 20 | 450 | 15 | 0 – 18 | 1045 |
| 5 | 324 | 395 | 109 | 3 - 83 | 1050 |
| 6 | 62 | 390 | 26 | 0 – 83 | 990 |
| 7 | 16 | 300 | 1 | - | - |
| 8 | 362 | 380 | 66 | 2 – 46.5 | 975 |
| 9 | 41 | 405 | 12 | 7 – 34 | 1020 |
| 10 | 17 | 305 | 2 | - | - |
| 11 | 23 | 420 | 9 | 3 - 40 | 970 |
| 12 | 18 | 320 | 1 | - | - |
| 13 | 5 | 335 | 4 | 4 - 28 | 1195 |

The higher population density of Vs profiles is within the GGD8 (Po River plain), particularly next the city of Torino, and along the Alpine River floors (GGD 5) where most of the settlements are located. In these GGDs 66 Vs profiles for GGD 8 and 109 for GGD 5 reached the seismic bedrock, considered as Vs higher than 800 m/s (Figures 4b and 5b). It worth's mention that in GGD 8 the distribution of profiles reaching the seismic bedrock is concentrated near the borders with the Alpine chain (Figure 4a). The Vs,z distribution of the non-bedrock layers was also computed for each profile (Figures 4c and 5c) together

with the resulting Vs,h according to NTC (2018), i.e. the depth h is the depth of the Seismic Bedrock, if this is reached within 30 m, otherwise it is 30m.  (Figures 4d and 5d). The Vs,z is indeed usually considered as a closer representation of the physics of the earthquake amplification along the soil profile than the Vs layered profile (Comina et al., 2022). This allowed also to obtain a representative median Vs,z profile for the different GGDs (together with its standard deviation). The two GGDs show relatively similar distributions of both Vs,z and Vs,h reflecting the similarities of the Cover Terrains within these domains (see

also later for more comments on this aspect).

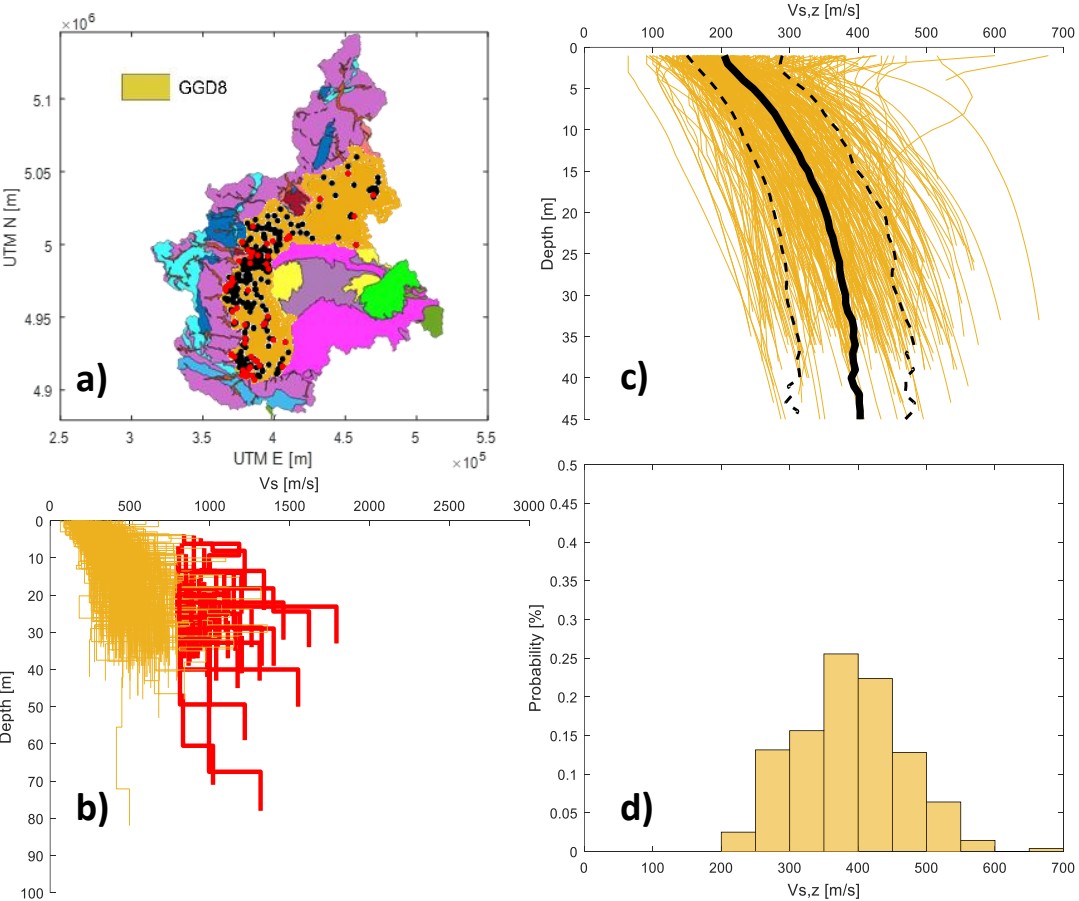

**Figure 4. The results of the analysis for GGD 8: a) Vs profiles geographical distribution (black dots) and Vs profiles reaching the seismic bedrock (red dots); b) All Vs profiles (orange lines) indicating bedrock velocities (red lines); c) Vs,z profiles for the Cover Terrains or weathered Geological Bedrock  and their mean (continuous black line) and**

**standard deviation (dashed black lines); d) Vs,h distribution following NTC (2018).**

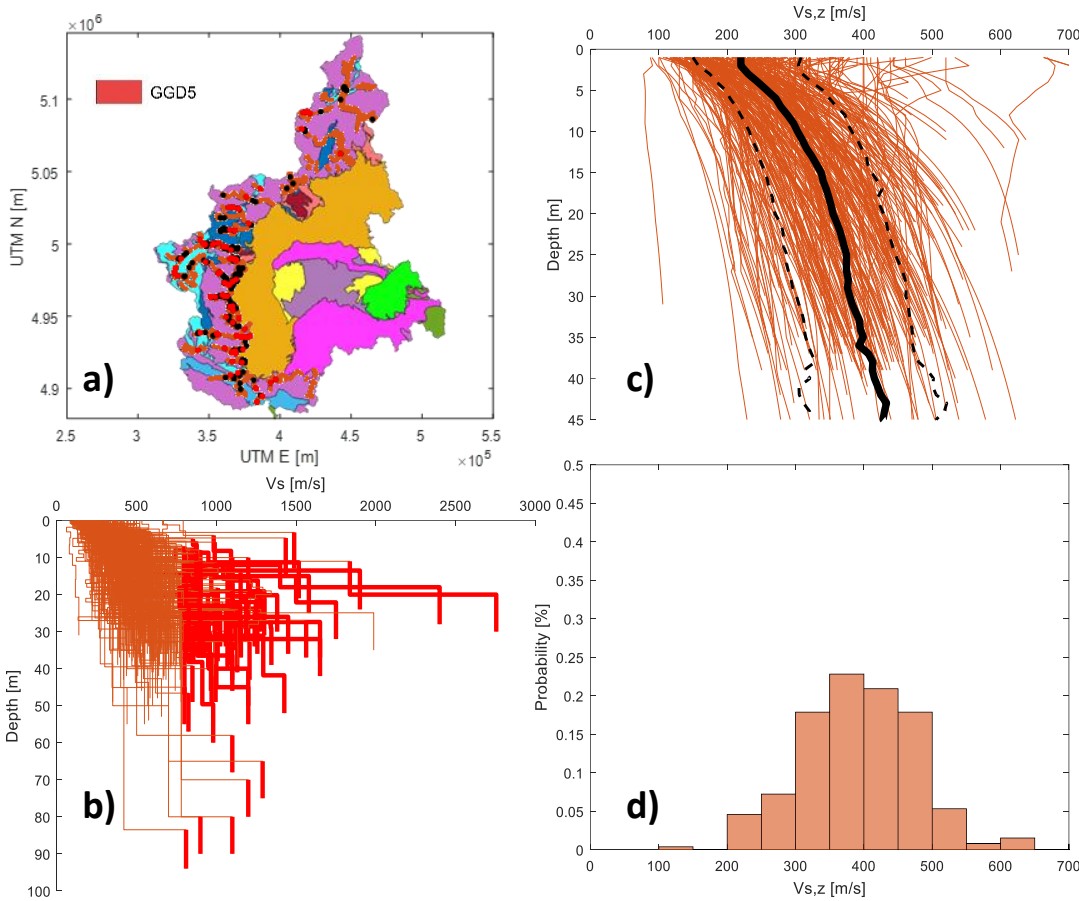

**Figure 5. The results of the analysis for GGD 5: a) Vs profile distribution (black dots) and Vs profiles reaching the seismic bedrock (red dots); b) All Vs profiles (orange lines) indicating bedrock velocities (red lines); c) Vs,z profiles for Cover Terrains or weathered Geological Bedrock and their median (continuous black line) and standard deviation (dashed black lines); d) Vs,h distribution following NTC (2018).**

Based on available data from all the GGDs, keeping in mind the representativeness of the results as a function of data coverage and distribution, the median Vs profile and its corresponding uncertainty as a function of depth was also calculated following the approach by Toro (2022). This calculation captures the central tendency of the profiles, evaluated at 1-meter intervals for each GGD, with uncertainty characterized by the logarithmic standard deviation ($\sigma_{lnV}$). These analyses were performed on all the GGDs with the exception of GGD 13 which contains very few Vs profiles and result therefore not statistically significant. The median Vs profiles, accompanied by uncertainty bands equal to ±1 standard deviation, highlight the differences between the different GGDs (Fig. 6).

When a high number of profiles are available and layer boundaries are not concentrated at specific depths (e.g., in geologically different settings), the median profile tends to be smooth (Toro 2022). Some median Vs profiles show larger uncertainty (e.g., GGD1, GGD2, GGD3, and GGD4) compared to others that exhibit lesser variability (e.g., GGD5, GGD7, GGD8, and

GGD12), regardless of the number of profiles available for the calculation. Specifically, profiles GGD5 and GGD8 show lower uncertainty and a smoother trend with depth, despite being derived from a higher number of profiles.

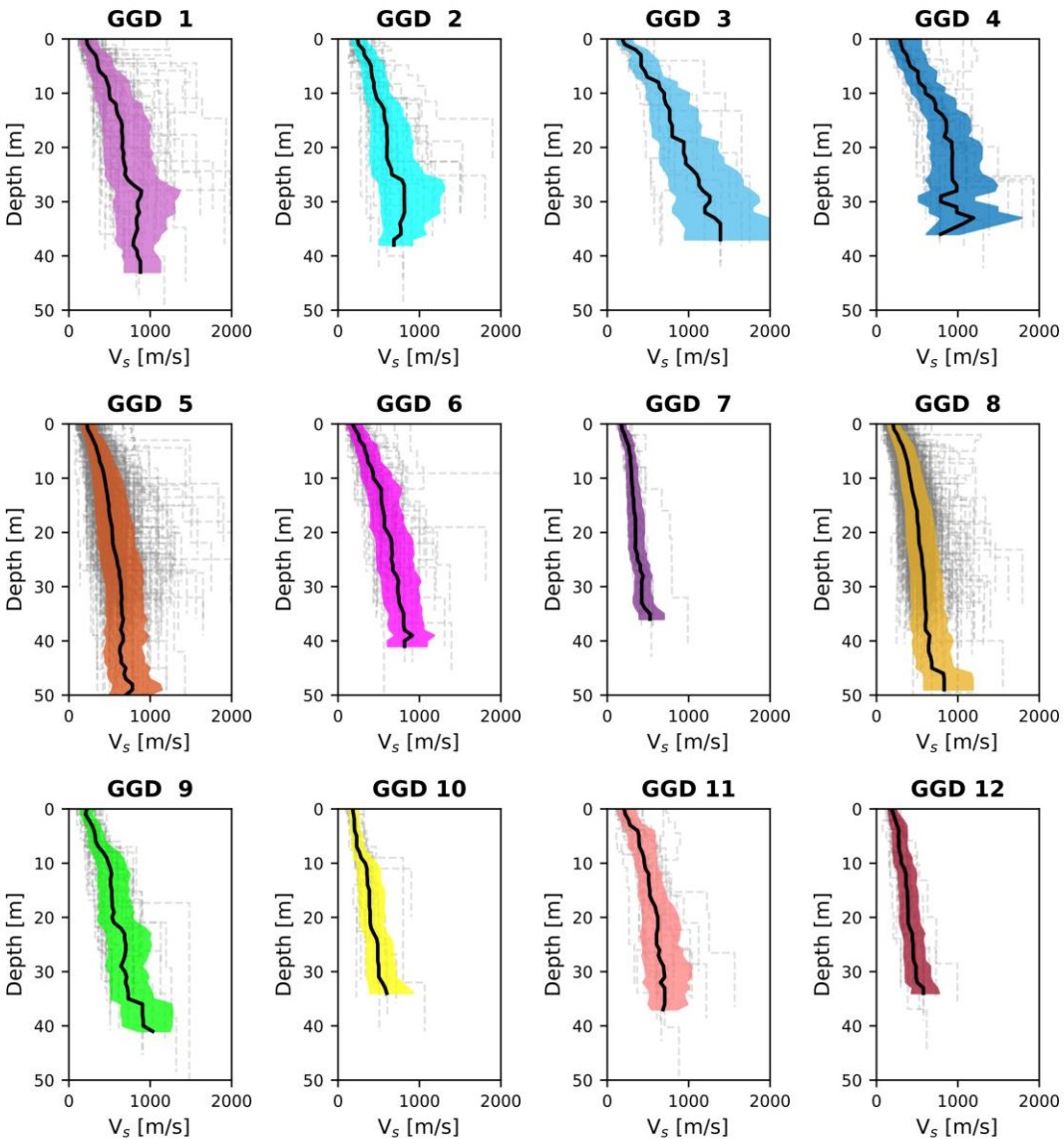

**Figure 6. Median Vs profiles (black line) with uncertainty bands of ±1 standard deviation (colored area) as σlnVs. For each panel, the GGD code is indicated, and the Vs profiles used to calculate the median are represented by grey dashed lines.**

Similar procedure was applied also in terms of median Vs,z profiles in the different GGDs (Figure 7). This analysis allows for the estimation of uncertainty and provides insight into the trend of the Vs,z profiles as a function of depth. Also, specific analyses of the median Vs,z profiles of the only cover terrains were performed. These last results are reported in terms of the

only median profiles in Figure 8 grouping all the GGDs (with the exception of GGD 13) to allow more specific comparison on the velocity distributions.

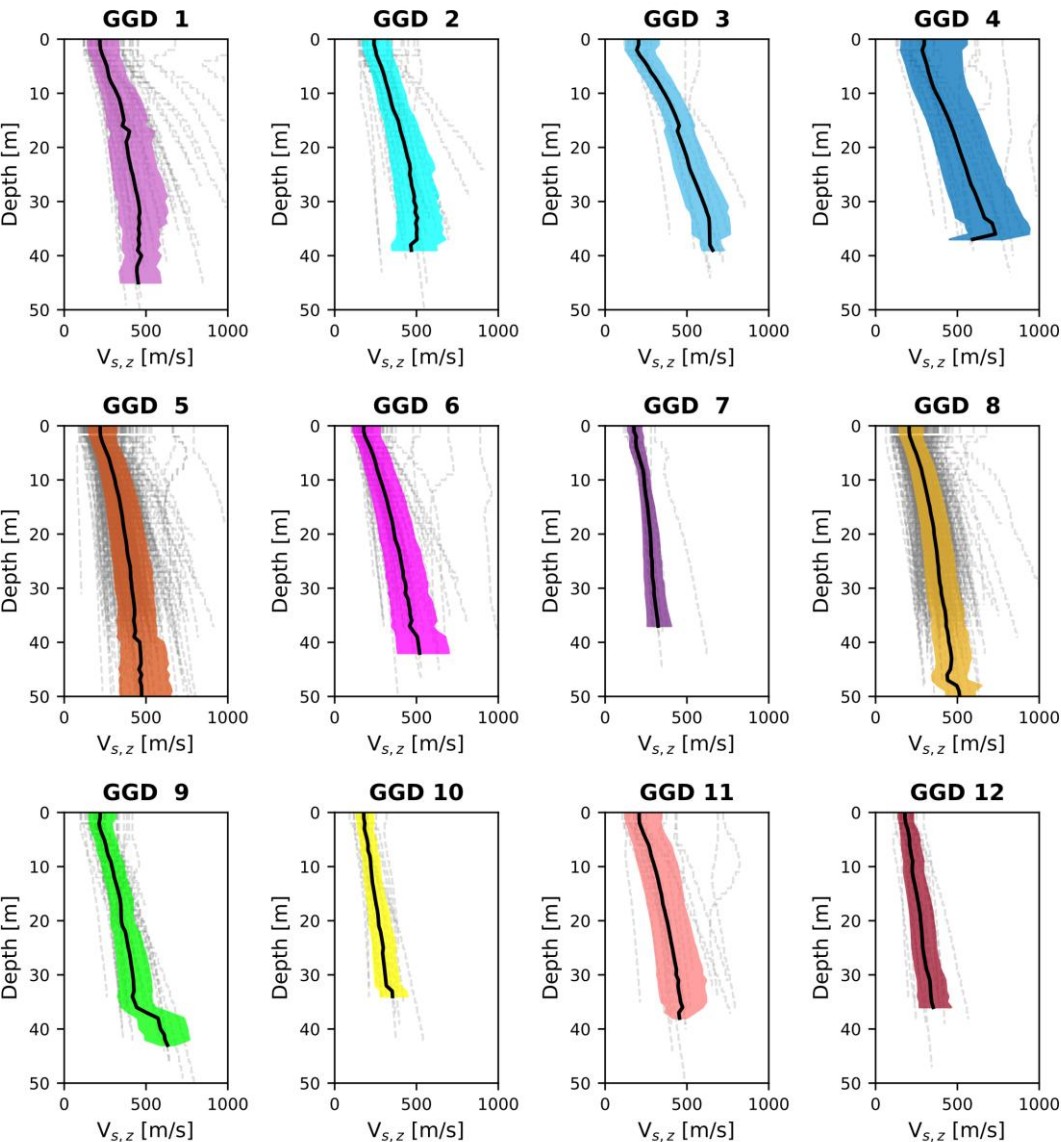

**Figure 7. The median Vs,z profiles are shown (black line), with uncertainty bands of ±1 standard deviation (colored area) representing σlnVs,z, for different GGDs. Each panel displays the GGD code, and the Vs,z profiles used to calculate the median are indicated by grey dashed lines.**

The distribution of the median Vs,z profiles show groups of GGDs with very similar behaviour, reflecting similar properties of the Cover Terrains. With this respect the different GGDs were initially selected only on the basis of the Geological information and the Vs distribution over the GGDs is considered as a second step of the research in order to check if geological

diversities correspond to different values of seismic velocity. Indeed, taking into account the uncertainties, the 13 proposed GGDs may be further simplified. In fact, the median Vs profile of one GGD may fall within the standard deviation boundaries of another (see Figure 7). Therefore, some GGDs may be grouped together, based on the Vs profile values and their uncertainty.

In particular, the GGDs' falling within the Alpine chain (GGDs 2 to 4) show higher Vs,z distributions with depth (see Figure 8), different to the others, due to the presence of very coarse-grained Cover Terrains, typically along the slopes (debris-slope, glacial, fluvio-glacial) or within the valley floors (alluvial, alluvial fan, glacial, fluvio-glacial), whose thickness is in the 3-100 m range. Conversely, GGD 7 (Hilly Pliocene Basins), 10 (Hills Rivers Plain) and 12 (Fluvio lacustrine) showed lower Vs,z distributions with depth (see Figure 8), reflecting the mainly fine-grained (sands, silts, clays and minor gravels) slope, alluvial

and lacustrine deposits, up to 50 m thick.

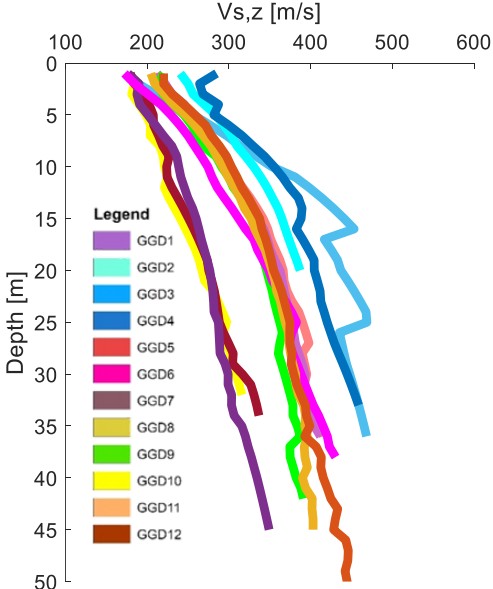

**Figure 8. Median Vs,z profiles for the Cover Terrains for each GGD.**

Following a global approach to the data analysis the median Vs,z profiles and their standard deviations, eventually merged between similar GGDs, could be adopted as the basis for randomization and amplification simulations within the Region or in

similar geological contexts. For this purpose specific randomization approaches, based on the same Vs,z (Passeri et al., 2020) or on usually adopted Vs randomization criteria (e.g. Toro, 2022) could be adopted using the data contained in the database as fundamental starting point. This proposed global approach allows to overcome the limitations inherited by the uncertainties of the specific litho-stratigraphic settings within each GGD, due to the regional scale of observations.

As an example, the approach proposed by Romagnoli et al. (2022), aimed at estimating a power law model of the trend of Vs

values with depth, was adopted. This model can be reported in linear terms following the form:

$$\ln(Vs) = b + a\ln(z) \qquad (1)$$

where z represents the depth and *a* and *b* are empirical parameters determined through the linear regression of Vs profiles. In particular, the term *b* represents the value of $\ln(Vs_1)$, the Vs at a depth of 1 m, while the term *a* modulates the gradient of Vs with depth. Through regression it is also possible to determine the standard deviation ($\sigma$) and determination coefficient ($R^2$) associated with this law, as well as the 95% confidence intervals ($\Delta a$ and $\Delta b$) associated with the estimate of *a* and *b*. These parameters can allow further judgement on the effectiveness of the merging of different GGDs in a unique group or on the real differences among the various GGDs. This approach was tested merging some GGDs which should share similar geological characteristics or Vs distributions: i) the merged group of GGDs' within the Alpine chain (i.e. 1 to 5); ii) the merged group of GGDs' within river plains (i.e. 8 and 9); iii) the merged group of GGDs 7 (Hilly Pliocene Basins) and 10 (Hills Rivers Plain) and iv) the GGD12 (Fluvio lacustrine). As it can be seen in the results of these analyses, reported in Table 5, both Alpine chain and river plains GGDs show increased values of $Vs_1$ and *b* parameters with respect to Pliocene, Oligo-Miocene and Fluvio lacustrine GGDs. As already commented above, these differences reflect the prevailing lithologies of the GGDs with higher velocities for coarse-grained Cover Terrain in the first case and mainly fine-grained Cover Terrain in the second case.

**Table 5. Analysed parameters for the power law Vs distribution among some example GGDs.**

| GGD | $Vs_1$ | b | a | Δb | Δa | σ | $R^2$ |
|---|---|---|---|---|---|---|---|
| 1 to 5 | 241 | 5.49 | 0.25 | 0.03 | 0.01 | 0.30 | 0.37 |
| 8 and 9 | 219 | 5.39 | 0.27 | 0.03 | 0.01 | 0.28 | 0.41 |
| 7 and 10 | 165 | 5.11 | 0.29 | 0.10 | 0.04 | 0.27 | 0.46 |
| 12 | 165 | 5.10 | 0.30 | 0.16 | 0.06 | 0.31 | 0.41 |

Moreover, the collected data allowed to produce maps of relevant seismic parameters at the regional scale. Maps deriving from geological/geotechnical and geophysical information contained in the database have been produced in Surfer (Golden Software, LLC) environment considering a uniform interpolation grid of about 2 km for all the data.

The map of the prevalent geological-technical properties of the Cover Terrains within the first 30 m (Figure 9) shows the distribution based on the stratigraphical logs. The distribution of the Cover Terrains properties matches the geological-geomorphological information adopted for the GGDs definition. Prevalent coarse-grained Cover Terrains (gravels, pebbles to boulders) are distributed along the Alpine domains (GGDs 1 to 4), within the Alpine valley floors (GGD 5) and within the Alpine and Apennine Alluvial plains (GGDs 8 and 9). Finer-grained Cover Terrains (i.e. sands, silts and clays) characterise the Oligo-Miocene and Pliocene Basins (GGDs 6 and 7) and Hills rivers plain (GGD 10). A comparison with respect to the distribution of subsoil properties for GGD 8 and GD 6, clearly reflecting what commented above, is reported in Figure 10.

For ground response evaluations this type of information is essential for the analysis of the subsoil nonlinear hysteretic behaviour. This is indeed usually described through appropriate shear modulus reduction and damping ratio curves. In common practice, in absence of specific laboratory tests, these curves can be estimated by employing empirical regression models (e.g. Vucetic and Dobry, 1991; Darendeli, 2001; Ciancimino et al. 2020; Wang and Stokoe, 2022), calibrated on large experimental datasets (e.g. Gaudiosi et al., 2023; Ciancimino et al., 2023), to correlate the soil physical properties, and their statistical distribution in the investigated units, with their nonlinear hysteretic behaviour. With this respect the information contained in the databased presented in this work can be adopted as key data for large-scale regional hazard assessments or in similar geological contexts.

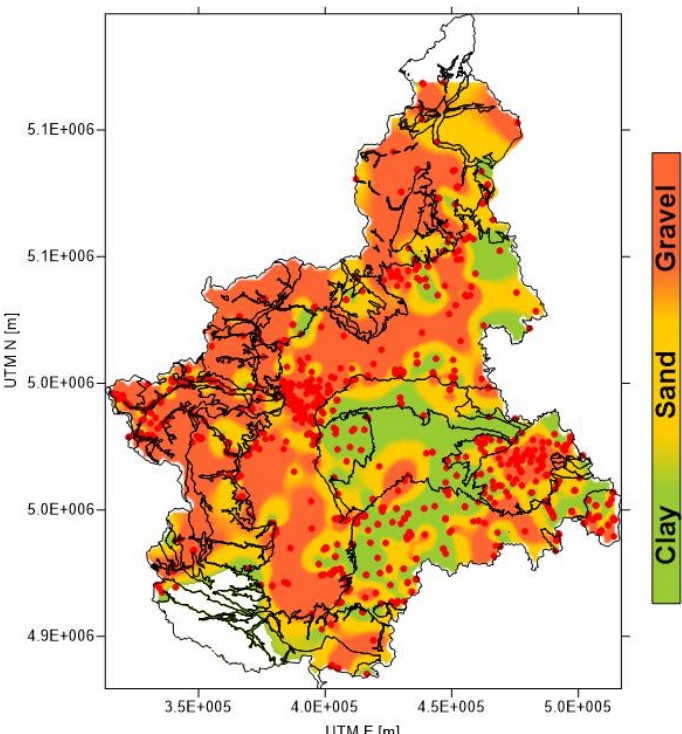

**Figure 9. Geological/geotechnical information derived by the stratigraphical logs database: map of the prevalent properties of the Cover Terrains within the first 30 m.**

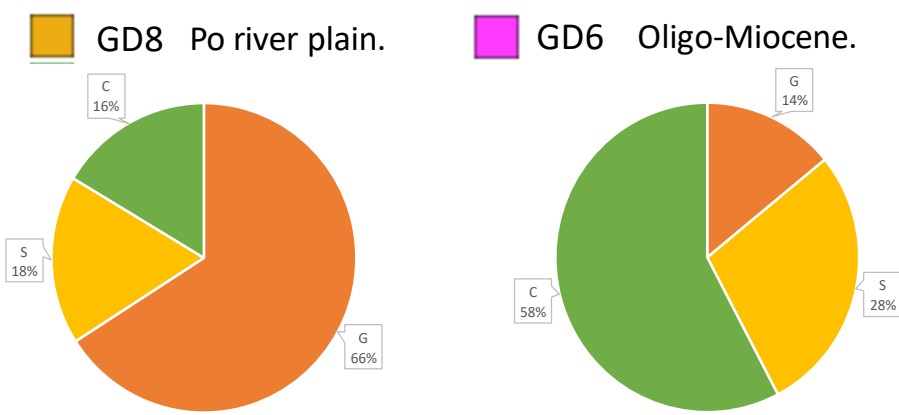


**Figure 10. Comparison on the distribution of Cover Terrains properties within the first 30 m for GGD 8, on the left, and GGD 6, on the right.**

A further essential complementary information for ground response analyses is the evaluation of the thickness of the Cover
Terrains, i.e. the Bedrock depth. With this respect, using the data from the database, the estimated Geological Bedrock depth distribution from stratigraphic logs (Figure 11) and the Seismic Bedrock depth distribution based on Vs profiles (Figure 12) have been compared.

The depth of the Geological Bedrock (Figure 11) shows a good correspondence with the attended Geological-Geomorphological setting. Increasing bedrock depths are observed along the main alluvial plains (i.e. GGD 8, Po River plain,
and GGD 9, Apennine plain) whereas the depth decreases along the Alpine chain and in the Oligo-Miocene (GGD 6) domain, that are those with higher relief energy and therefore more eroded landscapes. The only exception to this model is the presence of the Alpine valleys (GGD5) where the thickness of the valley floors deposits increases due to the glacial over-excavation. The same feature is observed within GGD8 (Po River plain) where local increase of depths characterises the tectonically downthrown buried structures (i.e. Savigliano Basin) and the buried continuation of the over-excavated glacial Alpine valleys
(Irace et al., 2009; Gianotti et al., 2015; Ivy-Ochs, et al., 2018).

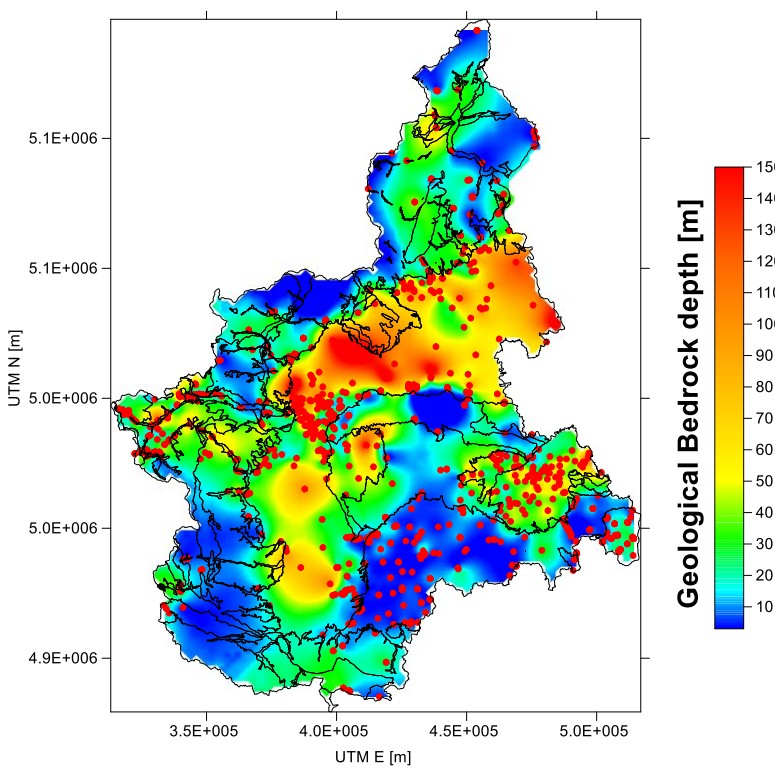

**Figure 11. Geological/geotechnical information presented in the database: map of the estimated Geological Bedrock depths.**

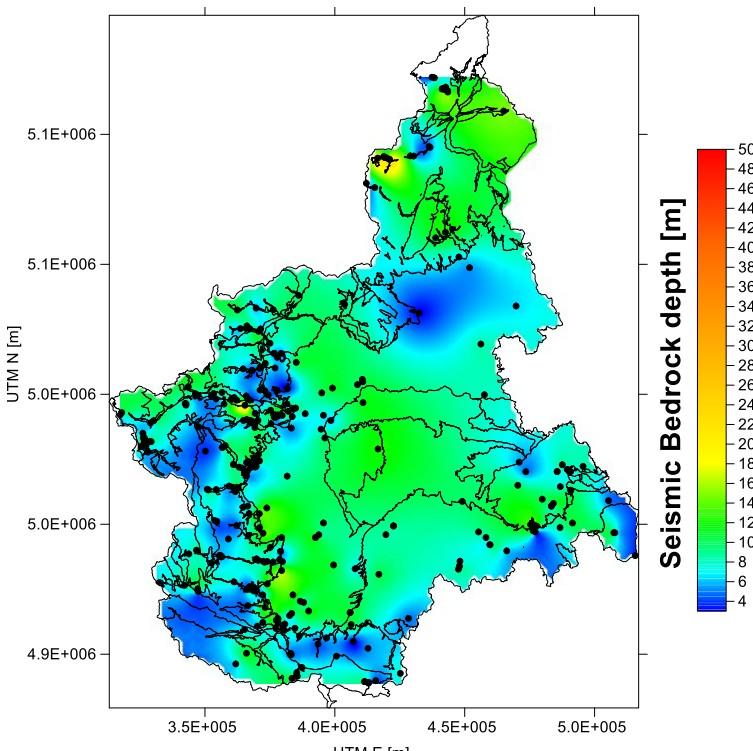

 **Figure 12. Geophysical information presented in the database: map of the estimated Seismic Bedrock depths.**

Comparing the Geological Bedrock depth (Figure 11) and the Seismic Bedrock depth (Figure 12) maps, the last is generally shallower than the former suggesting that the Vs profiles reach velocities of 800 m/s within the Cover Terrains, i.e. more packed or cemented or coarser-grained layers. This is highly relevant for ground response analyses since usually materials having this propagation velocity have a "rigid" behaviour. Nevertheless, the Seismic Bedrock map (Figure 12) still reports a setting coherent with the geological information, showing reduced bedrock depths along the Alpine chain. However, the number of data points in this last map is reduced with respect to the others (see also Table 4).

Finally, availability of Vs profiles in the presented database allowed also to represent at the regional scale the distribution of $V_{s,h}$ according to NTC (Figure 13) and of $V_{s,30}$ (Figure 14), where $V_{s,h}$ is the harmonic average shear-wave velocity down to the depth h of the Seismic Bedrock, if this is reached within 30 m, otherwise h is 30m, and $V_{s,30}$ is the harmonic average shear-wave velocity down to the depth of 30 m. The two maps show partially similar features. The $V_{s,h}$ map (Figure 13), according to its formulation, report generally lower velocities given that the Vs values reported pertain to the only cover deposits (i.e. Vs id evaluated only till the depth h of the seismic bedrock). Conversely the $V_{s,30}$ map (Figure 14) better represents the average increased values of Vs, generally above 500 m/s, within the Alpine Ridge GGDs where the shallower bedrock depths weights more on the velocity distribution. Notwithstanding this general difference, in both maps GGD 7 (Pliocene Basins) is characterised by generally lower velocities related to finer-grained Cover Terrains, as already evidenced

in the velocity distribution curves (see Figure 8). Similarly, GGD12 shows a clear localized velocity reduction in both maps contrasting with the coarser-grained and thick moraine deposits of GGD 11, bounding the same domain. Also partially high Vs,h and Vs,30 are observed within the GGD6, reflecting the presence of shallower geological and seismic bedrocks.

At the regional scale similar attempts to map the ground zones having a homogeneous seismic response (i.e. De Ferrari et al., 2015) and a similar map of Vs,30 (i.e. Perrone et al., 2015) have been already conducted in the past. With respect to these previous papers the present work is based on a significantly increased data coverage (stratigraphical and seismic) increasing the reliability of the regional view, also including the Geological-Geomorphological modelling as a constraint for any further analysis. It must be also underlined that with respect to the similar attempts mentioned the presented maps are only data driven,

i.e. developed without a specific geologically-based strategy (like in De Ferrari et ail., 2015) and the GDDs are in this respect only used for post interpretation. The confirmation of the coherence of the maps presented in the present study with the geological distribution is therefore a confirmation of the data quality and of the developed methodology.

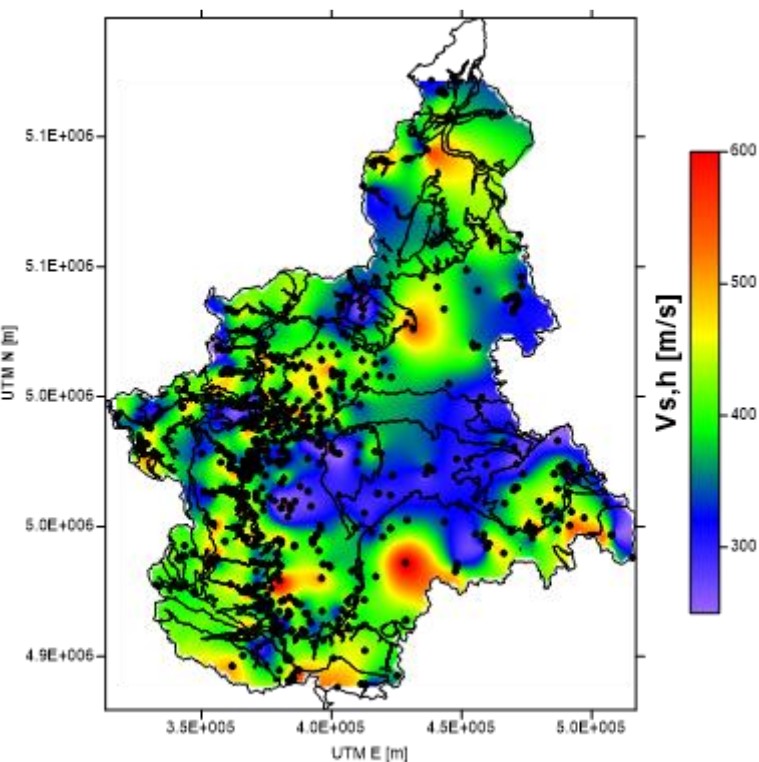

**Figure 13. Geophysical information presented in the database: map of the Vs,h distribution.**

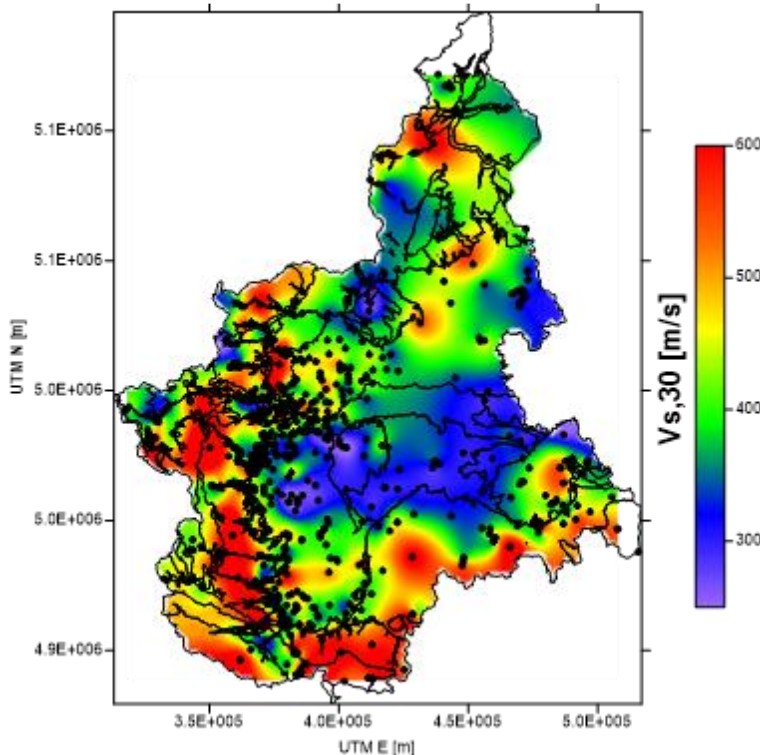


**Figure 14. Geophysical information presented in the database: map of the Vs,30 distribution.**

More generally, attempts to relate Vs,30 to Vs,z were evaluated in the literature: Boore (2004) used regression of data from boreholes in California to derive equations giving Vs,30 in terms of Vs,z. Other studies (e.g. Cadet and Duval, 2009) have used velocity profiles based on borehole measurements at KiKnet sites in Japan to derive similar relations. The data provided

in the database could be adopted for the verification of the above formulations or for similar type of analyses. Specifically, we analysed the Vs,30 to Vs,z data distribution contained in the database from z=5 to 20 m in depth and fitted the data following Boore et al. (2011) second-order polynomial relationship in the form:

$$\log(Vs, 30) = c_0 + c_1 \log(Vs, z) + c_2 \left(\log(Vs, z)\right)^2. \qquad (2)$$

Results of these analyses are reported in Figure 15 and Table 6. It can be evidenced that, as attended, the reliability of the correlation and its determination coefficient ($R^2$) increases with increasing depths. Also, shape of the correlations and their trend with increasing depth is coherent with what observed in Boore et al. (2011) for Japan, California, Turkey, and a mix of locations in Europe.

The database presented in this work will be the starting point for further work, i.e. numerical simulations of the seismic ground

response over statistically representative samples of the different GGDs in order to produce "amplification abacuses" for the quantification of local stratigraphic amplifications of the seismic ground motion over the Region.

**Table 6. Coefficients of Equation (2) in text, relating log Vs,30 to log Vs,z.**

| Depth z [m] | $c_0$ | $c_1$ | $c_2$ | $R^2$ |
|:---:|:---:|:---:|:---:|:---:|
| 5 | -0.6944 | 2.0493 | -0.2792 | 0.59 |
| 10 | -0.8801 | 1.9686 | -0.2245 | 0.77 |
| 15 | -0.3829 | 1.4217 | -0.0923 | 0.89 |
| 20 | -0.1669 | 1.1807 | -0.0365 | 0.96 |


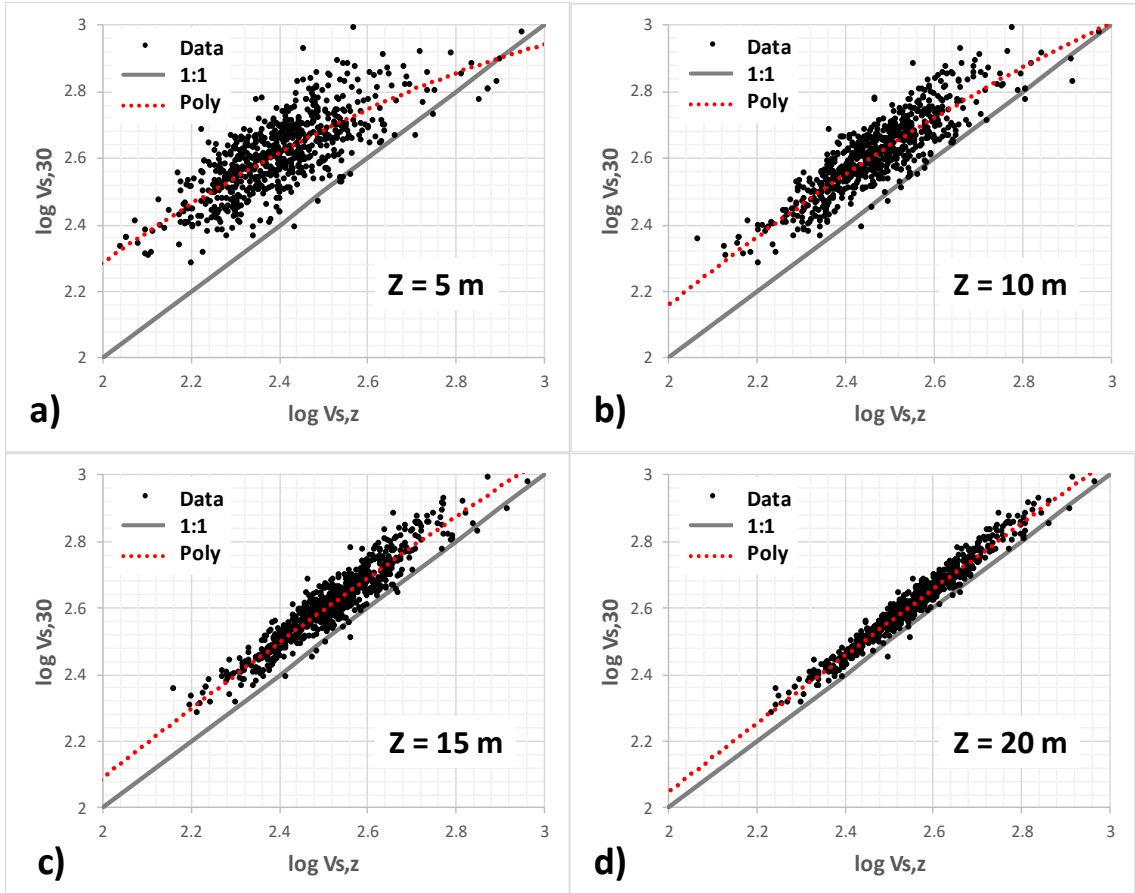

**Figure 15. Correlation of log Vs,30 and log Vs,z from the data contained in the database at different depths.**

## 5 Data availability

The database provides insights into the geological and geophysical features of the Piedmont region in Northwest Italy. It includes the Geological-Geomorphological Database, the Geotechnical Database, and the Geophysical Database. The database is referred to as Comina et al. (2025) and can be downloaded at https://doi.org/10.5281/zenodo.13685087.

## 6 Conclusions

In this work a new extensive database of Vs profiles and geological technical properties of the cover terrains over the Piedmont Region (NW Italy) has been shown and presented. The data are obtained through a specific workflow developed for their evaluation at the regional scale, merging geological information and specific geophysical data collection. Therefore, this paper: i) provide a new, extensive (i.e. containing more than 1000 profiles), database of Vs profiles to be used as the basis of randomization approaches also in different geological contexts; ii) provide discussion, from specific analyses, of median properties of the different investigated geological units to be eventually adopted with similar approaches as reference for similar materials in analogous geological contexts; and iv) provide examples, from specific analyses, of relevant parameters maps at the regional scale to be adopted with similar approaches for specific studies and or ground response regulations at the regional scale, iii) provide a workflow to be adopted for the same aim for evaluation of Vs profiles distribution at the regional scale even in different case studies.

## Author contributions

CC and PP designed the study. CC, PP, GMA, and CB performed the fieldwork. CC, PP, and GMA collected the data and assessed its quality. CC performed the analysis. CC, PP and GMA wrote the manuscript and created the figures. All authors contributed to the discussion and revision of the manuscript.

## Competing interests

The contact author has declared that none of the authors has any competing interests.

## Acknowledgments

Authors are indebted with the Sismic Sector of Piedmont Region for their support and data availability and with Techgea S.r.l. for the sharing of additional shear wave velocity profiles, fundamental to complement the data.

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
