# Peer review of "Regional scale shear wave velocity profiles for ground response analyses and uncertainties evaluations – the Piedmont Region (NW Italy) Database."

_Earth System Science Data, 2024_

## Author Response (AR1)

*Reviewer 1 comments:*

*The paper entitled "Regional scale shear wave velocity profiles for ground response analyses uncertainties evaluations – the Piedmont Region (NW Italy) Database" (by Comina et al.) presents a database of Vsz profiles in regional scale for use in seismic response analyses and an original methodology to create it. The paper deserves publication to the ESSD only if the following remarks are taken into account.*

*General remarks:*

*In the title I guess that an "and" is missing to make it more complehesible (e.g. "…for for ground response analyses and uncertainties evaluations…").*

A: The paper title was changed according to the suggestion.

*The first two lines of the Abstract are repeated in the first two lines of the Introduction.*

A: Thank you for noticing. We changed the first lines of the abstract in order to avoid repetition.

*Careful check of References is needed. For instance the NTC(2018) appears several times in the manuscript but it doesnot appear in the References.*

A: Thank you for noticing. We have included the missing reference and double checked all the other references to ensure that nothing is missing.

NTC, 2018. Norme Tecniche per le Costruzioni Consiglio Superiore dei Lavori Pubblici, Ministero per le Infrastrutture e dei Trasporti.

*Comparing the Vs,h map (Fig. 11) with the Vs,30 (Fig. 12) the authors stare that "The two maps show partially similar features". This expression seems quite vague and needs more clarification. For instance, it is suggested to the authors to use the relations proposed by Boore et al. (BSSA, 2011), This expression seems quite vague and needs more clarification that related globally Vs,30 values with Vs,h with satisfactory correlation. In this way, the author can cross-check their results and better interpret their findings.*

A: Thank you for your observation. Indeed, the comment related to the two maps was a little vague and need more elaboration. We have tried to provide a more in depth discussion on the comparison between the two maps. However, we also think that there is a partial misunderstanding on the maps presented: the Vs,h data discussed here are not the same as Vs,z data discussed by Boore et al. (2011) but are related to the NTC 2018 formulation, i.e. the depth h is the depth of the Seismic Bedrock, if this is reached within 30 m, otherwise it is 30m. This was also already stated in the text.
Therefore, the correlations proposed by Boore et al. (2011) are not directly applicable to the data contained in the two maps. We have also some conflicting opinion on the application of these correlations, given that extrapolation to Vs,30 of Vs,z data obtained at shallower depths is, to our judgement, not completely correct. Nevertheless, we met the suggestion of the reviewer and proposed some additional data elaboration in line with what contained in the Boore et al. (2011) paper and we referenced this last paper in the text. This kind of application is in our opinion coherent with the purposes with which we have developed the database and with the use that the scientific community can make of it.

*In Figures 4c and 5c large dispersion of the Vs,h profiles is observed. This means that if uncertainties are taken into account in ground response analyses, discrimination between the 13 proposed categories may be meaningless. That is, some categories may fall in the zone of +-one standard deviation of another. To clarify this issue, a quantitative parametric analysis is needed, including ground response analyses of the average +- one s.d. of the Vs,h profiles presented in Figure 6. Such a parametric analysis can improve the validity of the created database and encourage its use in regional scale.*

A: Thank you for your observation. We completely agree that not all the 13 categories are necessary for describing the geological setting of the Region and that there is a significant superposition among the different evidenced GGDs. Therefore, a merging of some domains is for sure recommended. The Geological Domains were selected only on the basis of geological information and the Vs distribution considered as a second step of the research, in order to verify if geological differences reflect also in seismic velocity differences. This is not always the case, as an example GD 1 to 5, all pertaining to the alpine setting, could be easily merged together in a unique domain, and this is what indeed we performed in further elaborations of the database. Notwithstanding the above considerations, that we have added to the text of the paper as commenting material, we would like to keep the database subdivision in 13 domains as it is at present. We indeed think that different applications of the information contained in each GGD could be still possible depending on the intended use of the data.

To quantify the uncertainty in Vs profiles, we utilized data from the GGDs to compute the median of Vs and its corresponding uncertainty as a function of depth, following the methodology outlined in Toro (2022). This approach calculates the central tendency of the profiles, evaluated at 1-meter intervals for each GGD, with uncertainty represented by the logarithmic standard deviation ($\sigma$ lnVs). Building on this methodology, we extended the analysis to Vs,z profiles, applying the same approach. This allows for a more comprehensive understanding of variability across different GGDs.

To support this analysis, we have added two new figures to the manuscript, which visually present the median of Vs and Vs,z profiles along with their uncertainty bands. For each GGD, the median profile serves as the baseline. In these figures, we display all the median profiles, accompanied by uncertainty bands of $\pm 1$ standard deviation, to highlight the variability between the different GGDs. Additionally, we have incorporated new sections of text to describe the results of this analysis, emphasizing the differences observed between the Vs and Vs,z profiles across the GGDs. This comparison provides a clearer understanding of how uncertainty with depth influences the variability of the profiles, thereby enhancing the insights presented in the manuscript.

As also mentioned by reviewer 2, further data elaboration on the presented database, is outside the scopes of the paper and of the aim of the journal. Here we make available the database, the use that each researcher would make of it is open. With this respect we do not think it that is necessary here to perform all the suggested ground response analyses. Moreover, ground response analyses, as commented in the text, would require additional parameters and additional uncertainty sources (i.e. G and d curves) which, in our opinion, would not necessarily help clarify the Vs and Vs,z variability the reviewer aims to address. In our view, this extends beyond the scope of the work and is not directly related to its primary objectives.

*Minor remarks*
*In the header of Table 4: an "S" is missing (e.g. Seismic bedrock depth range[km])*
*In the caption of Fig. 4d: "…following NTC(2018)"*
*In line 213: "…similar behaviour, …"*

A: We have corrected the suggested minor remarks and further minor mistakes evidenced from a rereading of the whole paper.

*Reviewer 2 comments:*

*I have found the paper of potential interest for the essd reader and for a wider community of seismologists and engineers involved in seismic hazard assessment. The scientific background is sound and the whole workflow is satisfactorily illustrated. Results are significant and the database provides an important contribution for the study of site effects at regional scale. However, important information is apparently lacking and this prevents the possible direct application of outcomes from this study.*

*The randomization process to be applied to account for uncertainty in the Vs soil profiles when site response is estimated requires a quantitative formalization (average and standard deviation of Vs values as a function of depth, correlation between subsequent layers) which is lacking. This does not allow evaluating significance of differences of the Vs profiles in Figure 6. This formalization was possibly beyond the scope of a paper aiming at providing the database, but also represents an important limitation of the work. Supplying this information could largely improve the impact of the paper.*

A: We conducted an analysis of the uncertainty in the Vs and Vs,z profiles for a comparison between the different GGDs. To support this, we have added two new figures to the manuscript, visually presenting the median profiles and their uncertainty bands. The analysis highlights the differences observed across the GGDs, providing a clearer understanding of how depth-related uncertainty influences the variability of the profiles.

As for the response to a similar comment from reviewer 1 we completely agree that not all the 13 categories are necessary for describing the geological setting of the Region and that there is a significant superposition among the different evidenced GGDs. Therefore, a merging of some domains is for sure recommended. With this respect we have added some considerations in the text and we have also suggested possible formulation for the randomization process, in line with the reviewer observation.

However, as already commented in the paper, alternative randomization approaches are possible on the data that we are making available in the paper and we agree with the reviewer that a precise formalization is outside the scopes of the paper. What discussed and suggested here is only a possible alternative but further and different formulations could be possible.